# Matcha Green Tea: Chemical Composition, Phenolic Acids, Caffeine and Fatty Acid Profile

**DOI:** 10.3390/foods13081167

**Published:** 2024-04-11

**Authors:** Joanna Kika, Karolina Jakubczyk, Alicja Ligenza, Dominika Maciejewska-Markiewicz, Kinga Szymczykowska, Katarzyna Janda-Milczarek

**Affiliations:** Department of Human Nutrition and Metabolomics, Pomeranian Medical University in Szczecin, 24 Broniewskiego Street, 71-460 Szczecin, Poland; joanna.kika@pum.edu.pl (J.K.); dominika.maciejewska@pum.edu.pl (D.M.-M.); kinga.szymczykowska@pum.edu.pl (K.S.); katarzyna.janda.milczarek@pum.edu.pl (K.J.-M.)

**Keywords:** matcha tea, green tea, nutritional composition, fibre, protein, fatty acids, phenolic compounds

## Abstract

Matcha—Japanese powdered tea—is a variety of green tea (*Camellia sinensis* L.), one of the most popular beverages in the world. Due to the unique way it is grown, it contains high concentrations of health-promoting phytochemicals. The aim of this study was to determine the basic nutritional and phenolic composition of dry matcha green tea powder. The fibre content was determined according to the enzymatic–gravimetric method. Crude protein was measured by the Kjeldahl method. The total fat content was measured by the Soxhlet method, and the fatty acid profile was defined by the GC method. The determination of the phenolic acid and caffeine content was performed using high-performance liquid chromatography. The total fibre content of matcha was 56.1 g/100 g, with 52.8 g/100 g (94.1% of total fibre) of insoluble dietary fibre and 3.3 g/100 g of soluble fibre (5.9% of total fibre). The total protein content was 17.3 g/100 g. The total fat content in dry matcha was 7.285 g/100 g, comprising varying proportions of individual fatty acids, the highest ones being those of linoleic acid and α-linolenic acid. The caffeine content of matcha tea was 2213.492 µg/g. With regard to phenolic acids, the highest content was estimated for gallic acid (252.3755 µg/g). Matcha showed a particularly high content of total dietary fibre, with a predominance of the insoluble fraction. Matcha was found to be a valuable source of plant protein and unsaturated fatty acids, mostly of the omega-3 fatty acid family.

## 1. Introduction

Tea is one of the most popular liquids consumed by humans [1]. Due to its distinctive taste and aroma and numerous health-promoting benefits, as well as to socio-cultural factors, it is one of the world’s favourite beverages [2,3]. Matcha is a variety of Tencha green tea [4], which is gaining popularity on a global scale [5]. Unlike other types of tea, matcha is consumed in the form of powder containing all parts of the leaf [6,7]. It is particularly rich in antioxidant compounds due to its unique, traditional cultivation method [8,9]. This involves shading the tea plants under canopies for about 3 weeks before harvest [7,9,10]. Protection from the sun enables the plant to produce a high amount of bioactive compounds. The shading treatment alters its chemical composition, with an increase in theanine and caffeine content and a decrease in catechins, resulting in a higher proportion of the compounds responsible for the ‘umami’ taste. This is in contrast to the growth in full sun used for other teas, whose bitterness is enhanced by the catechin content [11,12]. Shading is also responsible for the unique flavour and the intense green colour of matcha. For this reason, matcha is highly prized for its quality and considered to be the most aromatic green tea [4,10,13,14]. The health-promoting properties of green tea and its chemical composition depend on the type of leaves used, the form of the leaves (powdered, loose leaf or bagged) and the method of tea preparation (brewing temperature, brewing time) [7,15].

Antioxidants are the main active compounds in green tea responsible for its health benefits [16], due to their antioxidant, antiviral and anti-inflammatory effects, stimulating immune and detoxification processes. Of particular note are polyphenols, a broad group of compounds including flavonoids and phenolic acids, which account for up to 30% of the dry weight [2,17]. Green tea consumption is associated with a reduced risk of many diseases, including cancer, by delaying the onset of risk factors associated with disease development [18,19,20]. The biochemical composition, and therefore the health-promoting properties, depend on the origin of the plant, the duration of shading from sunlight, the strength of the infusion and the brewing conditions, such as temperature and time [21].

Increasing knowledge of the chemical composition and biological properties of matcha tea infusions is contributing to its growing appeal to consumers, the food industry and the nutritional supplement sector alike. It has a low calorie content, besides being vegetarian and vegan. In addition, the visual appeal and the valuable health benefits of matcha tea fit in with the current trend of preferring natural and healthy foods. As well as being consumed in its typical infusion form, matcha tea is sold as a powder for direct consumption. The tea powder is increasingly used in the confectionery sector, as a food colouring and in capsule form [22,23]. Despite many studies on the composition and properties of green tea powder infusions, there is still a lack of data on the powder itself and its basic nutritional composition, including dietary fibre fractions and lipid profile. Therefore, the aim of the present study was to determine the basic nutritional composition of matcha green tea powder.

Our work focuses on the examination of tea powder, due to its increasingly widespread use in food processing and manufacturing. In addition, our work is one of the first to identify specific values for the insoluble and soluble fraction of dietary fibre in matcha tea, as well as the content of various phenolic compounds, which other researchers did not address. The reports derived from our study complement the existing results, providing a full description of Matcha tea in terms of its chemical and nutritional value.

## 2. Materials and Methods

### 2.1. Materials

The following reagents, consumables and special materials were used in this study: ethanol (96%, 70%) (Chempur, Piekary Śląskie, Poland), petroleum ether ACS grade (Chempur, Piekary Śląskie, Poland), Total Dietary Fiber Assay Kit (Megazyme, Bray, Ireland), Tashiro indicator (Chempur, Piekary Śląskie, Poland), hydrochloric acid 35–38% (Chempur, Piekary Śląskie, Poland), sodium hydroxide 0.1 M (Chempur, Piekary Śląskie, Poland), MES-2-(N-morpholio)ethanesulfonic acid (Chempur, Piekary Śląskie, Poland), tris(hydroxymethyl)aminomethane 99% (Chempur, Piekary Śląskie, Poland), acetone ACS grade (Chempur, Piekary Śląskie, Poland), NaOH ACS grade (Chempur, Piekary Śląskie, Poland), chloroform ACS grade, BHT (butylated hydroxytoluene) (Chempur, Piekary Śląskie, Poland), KOH ACS grade (Chempur, Piekary Śląskie, Poland), methanol, HPLC grade (Chempur, Piekary Śląskie, Poland), boron trifluoride 14% in methanol (Merck, Darmstadt, Germany), n-hexane HPLC 99% (Chempur, Piekary Śląskie, Poland), capillary column of dimensions 15 m × 0.10 mm, 0.10 μm (Supelcowax™ 10 Capillary GC Column, Supelco, Bellefonte, PA, USA), internal standard heneicosanoic acid (Merck, Darmstadt, Germany).

### 2.2. Plant Material

The material tested was a high-quality organic Japanese matcha green tea (*Camellia sinensis*) powder, made from Tencha leaves, from the Uji region of Japan in Kyoto Prefecture. The tested tea was from the second and third harvests. The product was supplied directly by the distributor, but the purchase was made independently. Material from three independent packages from different production runs were included in the study. The hermetically sealed powder was stored at −4 degrees Celsius.

### 2.3. Basic Nutritional Composition Analysis

Crude protein (N × 6.25) was measured by the Kjeldahl method, in a Büchi distillation unit B−324 (Büchi Labortechnik AG, Flawil, Switzerland). The Kjeldahl method was used as described by the AOAC [24,25]. Samples of 0.5 g of raw material were weighed in flasks. Then, 15 mL of concentrated sulfuric acid and catalyst was added, and the mixture was mineralised (Buchi mineraliser). The mineralised sample was alkalised with 40% sodium hydroxide to convert the ammonium salt to ammonia by distillation in a KjelFlex K-360 unit (Buchi). The separated ammonia was determined by reaction with hydrochloric acid (0.1 M) (Chempur, Poland) and titration with sodium hydroxide (0.1 M) in the presence of a Tashiro indicator. A conversion factor of 6.25 was used to convert nitrogen to total protein.

The total fat content was determined using the methods described by the AOAC [24]. Total fat was measured by the Soxhlet method described in Polish standards, using petroleum ether [24]. Samples of 4.5 g of raw material were weighed into a thimble. The flask with the solvent (petroleum ether) was connected to the Soxhlet apparatus, and a backflow cooler was installed from above. The solvent was brought to a boil. The vapours entered the cooler, condensed and then filled the part of the apparatus containing the sample to be extracted. The solvent automatically flowed back into the flask at the bottom when a certain level was reached. The extraction time was 6 h. The solvent was removed by drying at 130 °C to a constant weight. The percentage of fat content was calculated from the dry weight.

The total fibre content was determined using methods described by the AOAC [24]. Total, insoluble and soluble dietary fibre was determined according to the enzymatic–gravimetric method, with α-amylase, protease and amyloglucosidase from the K-TDFR 01/05 kit (Megazyme International, Wicklow, Ireland) using a Fibertec 1023 system (Tecator Tech., Hoganas, Sweden).

### 2.4. Isolation of Fatty Acids

The matcha tea samples were prepared using an ultrasonic cleaner. The lipid layer from matcha powder was isolated by adding 3 mL of Folch mixture, followed by 100 µL of internal standard and 100 µL of BHT. The samples were incubated for 15 min at room temperature. The lipid fraction was transferred to a clean test tube, and the fatty acid esters were saponified with 1 mL of 2 mol/L of a KOH methanolic solution at room temperature for 20 min. Then, 2 mL of 14% boron trifluoride in methanol was added, and the mixture was incubated under the same conditions. In the final step, the obtained methylene esters were extracted, and then 2 mL of n-hexane and 10 mL of saturated NaCl solution were added.

After the organic phase was completely separated from water, one millilitre of the n-hexane phase was collected. The samples were stored at −80 °C until chromatographic analysis.

### 2.5. Chromatographic Analysis

The chromatographic analyses of fatty acids were performed using a gas chromatograph (Agilent Technologies 7890A GC System, Santa Clara, CA, USA) with an FID detector. In addition, a capillary column of dimensions 15 m × 0.10 mm, 0.10 μm (Supelcowax™ 10 Capillary GC Column, Supelco, Bellefonte, PA, USA) was used. The chromatographic conditions were as follows: the initial temperature was 60 °C for 0 min, it was increased at a rate of 40 °C/min to 160 °C (0 min), then at a rate of 30 °C/min to 190 °C (0.5 min) and finally at a rate of 30 °C/min to 230 °C in 2.6 min, at which it was maintained for 4.9 min. The final temperature was sustained for 4.9 min. The total analysis time was approximately 8 min. The carrier gas was hydrogen at a flow rate of 0.8 mL/min.

Fatty acids were identified by comparing their retention times with those of commercially available standards. ChemStation B.04.01 Software (Agilent Technologies, Cheadle, UK) was used for the quantitative analysis. The quantity of each acid was calculated relative to the internal standard, which was heneicosanoic acid (C21:0).

### 2.6. Determination of the Phenolic Acid and Caffeine Content

Liquid chromatography (Agilent Technologies 1260 HPLC System, ChemStation B.04.03 Software, Santa Clara, CA, USA) was used to determine the polyphenolic compounds. A Hypersil Gold column (150 × 4.6) with temperature maintained at 25 °C was used. The phenolic compounds were detected by UV absorption at λ = 278 nm. For the identification of each compound, retention times and comparison with standards under the same conditions were used. The mobile phase was composed of 1% of an aqueous acetic acid solution (A) and 100% of MeOH (B). The following gradient was used to elute the samples: 90% of A and 10% of B from 0 to 6 min, 84% of A and 16% of B from 7 to 25 min, 72% of A and 28% of B from 26 to 37 min, 65% of A and 35% of B from 38 to 47 min, 50% of A and 50% of B from 48 to 64 min and 90% of A and 10% of B from 65 to 70 min to recover the initial conditions before injecting a new sample. The flow rate was 0.8 mL/min, and the injection volume was 30 µL.

### 2.7. Statistical Analysis

Three samples were analysed in all experiments, and all tests were conducted at least in triplicate. Statistical analysis was conducted using Stat Soft Statistica 13.0 and Microsoft Excel 2017, and the results are presented as mean values and standard deviation (SD). The value distributions for each parameter were analysed using the Shapiro–Wilk test. The Kruskal–Wallis test was used to evaluate the differences between the studied parameters. The Spearman’s R test was applied to calculate the correlation coefficient. Differences were considered significant at *p* < 0.05.

## 3. Results

In terms of basic chemical composition, the characteristics of powdered Japanese matcha tea are presented in Table 1.

Matcha has a high content of dietary fibre, especially the insoluble fraction. The total fibre (TF) content of matcha was 56.1 ± 2.753 g/100 g, with 52.8 ± 2.611 g/100 g of insoluble dietary fibre (IDF) and 3.3 ± 0.141 g/100 g of soluble fibre. The percentages of the dietary fibre fractions are presented in Figure 1.

Matcha powder is also a good and underestimated source of plant protein, whose content was found to be 17.3 g/100 g, with 3 g/100 g of nitrogen. A conversion factor of 6.25 was used to convert nitrogen to total protein. Consequently, one tablespoon of dry matcha powder (approximately 12 g) contains 2 g of protein.

The total fat content of dry matcha powder was 7.285 g/100 g. The proportion of unsaturated fatty acids in total fat was 83.255%, with the remaining 16.745% being saturated fatty acids.

Matcha powder was found to contain unsaturated fatty acids from several omega families, with 0.206% of palmitoleic acid from the omega-7 family; 4.573% of oleic acid from the omega-9 family; 0.502% of vaccenic acid from the omega-7 family; 12.562% of linoleic acid from the omega-16 family; and 65.412% of α-linolenic acid from the omega-3 family. The lipid profile of matcha tea is shown in Table 2.

Of the unsaturated fatty acids, the omega-3 family was the most abundant omega family. The next most abundant family was the omega-6 family, followed by the omega-9 one, and the least abundant was the omega-7 family. Figure 2 shows the content of each omega fatty acid family in relation to the total fat content.

Two types of saturated fatty acids were found in the lipid profile. The content of saturated fatty acids was 15.4% for palmitic acid and 1.3% for stearic acid, together representing 16.7% of total fat. The detailed results are shown in Table 3.

During the determination of the phenolic compounds, a number of substances were detected. The most abundant compound in matcha tea was caffeine—2213.492 µg/g—while among the phenolic acids, gallic acid had the highest content—252.3755 µg/g. In the order from the highest to the lowest content, caffeine, gallic acid, epicatechin gallate, myricetin, 4-hydroxybenzoic acid, rutin, sinapic acid, ferulic acid, resveratrol, ellagic acid, apigenin, quercetin, p-coumaric and caempferol were determined in the material studied. Meanwhile, chlorogenic acid, caffeic acid, dihydroxybenzoic acid and 2-hydroxycinnamic acid were not detected in matcha tea powder. The detailed results are shown in Table 4.

## 4. Discussion

Matcha, a powdered Japanese green tea (*Camellia sinensis*), is widely popular worldwide, and its consumption continues to increase [26]. Depending on the form of consumption of matcha tea, different compounds and nutrients are expected to be released. Matcha tea is increasingly being consumed in the form of a food supplement, rather than just as an infusion [22,23,27]. Due to its unique cultivation method, it is rich in phytochemicals and offers many health benefits, having, e.g., powerful antioxidant properties. About 60–70% of the nutrients in green tea are insoluble, e.g., fat-soluble vitamins, insoluble dietary fibre, chlorophylls and proteins. On the other hand, 30–40% of them are soluble components, including polyphenols, water-soluble vitamins, caffeine, water-soluble dietary fibre, amino acids, saponins and minerals [22,28]. Despite the growing body of knowledge about the composition and properties of green teas, there is still a lack of research dedicated to matcha. Therefore, in this study, an attempt was made to conduct a detailed analysis of the nutritional composition of matcha tea powder.

A high intake of dietary fibre is associated with a reduced risk of many medical conditions, including cardiovascular disease, diabetes, high blood pressure, overweight, obesity and gastrointestinal disorders [14]. Additional benefits of fibre consumption include improved bowel function, fermentability by the gut microbiota and attenuation of the blood glucose and cholesterol levels [29]. The physiological effects of dietary fibre can be attributed to its chemical and physical qualities, such as viscosity, degradability, molecular weight, particle size, cation exchange properties and water retention capacity [30,31]. There is a strong correlation between the prevalence of chronic diseases, the health status of the population and the consumption of fibre-rich cereal products. Unfortunately, it appears that a large percentage of the population finds it difficult to meet even the minimum recommended fibre intake [32]. The insufficient supply of fibre in the diet has led to the growing popularity of fibre supplements and the fibre fortification of bakery products, confectionery, smoothies, cakes and dairy products. In light of the above, it is vital to identify high-fibre, unprocessed foods that are appealing to consumers.

In this study, the total fibre (TF) content of matcha was shown to be as high as 56.1 g/100 g. Dietary fibre has been recognised for many years as an essential component of a healthy diet. The recommended daily intake is 25–35 g [33,34,35]. One tablespoon (approximately 12 g) of matcha powder contains 6.73 g of dietary fibre, that is, 26.92% of the recommended daily allowance (RDA) of this nutrient for an adult. Consuming 3.4–4 tablespoons per day will meet the daily quota. In comparison, the fibre content of wheat bran is 50 g/100 g of product, that of wheat germ is 15.5 g/100 g, and that of linseed is 27.3 g/100 g. The data presented above suggest that matcha tea is as good a source of dietary fibre as other commonly consumed products.

There are few studies showing green tea to be a very good source of dietary fibre, especially of the insoluble fraction. These few studies reported a similar total fibre content in matcha, but the fibre fractions have not been investigated to date. Our study is the first to examine them. We investigated the content of insoluble dietary fibre in matcha, which was found to be 52.8 g/100 g (94.1% TDF). The content of soluble dietary fibre in matcha was 3.3 g/100 g (5.9% TDF). The effects of different types of dietary fibre vary, so a varied diet is recommended to provide all of them [30]. Different fractions and amounts of dietary fibre in a meal can affect gastric emptying rate, intestinal transit time and nutrient absorption. By the time fibre reaches the colon, it is at least partially fermented and interacts with the gut microbiome [36,37,38]. Insoluble fibre has a number of properties: it stimulates the intestinal peristalsis, shortens the intestinal transit time, increases the stool volume [39], stimulates the blood supply to the intestine [40], suppresses appetite, prolongs post-prandial satiety [29], prevents and helps treat constipation and prevents colon and colorectal cancer [41]. Soluble fibre has gelling properties and cation exchange capacity, prevents colon and colorectal cancer [41], reduces blood cholesterol levels, is additionally converted to short-chain fatty acids [39], prolongs satiety and suppresses appetite [42]. The total dietary fibre intake was shown to be associated with a reduced risk of cardiovascular disease [39]. Thus, the properties of matcha can be mainly attributed to the predominance of insoluble fibre, which has the added benefit of providing food for intestinal bacteria.

In our study, the total protein content of matcha was 17.3 g/100 g, and the nitrogen content was 3 g/100 g. In comparison, similar amounts can be found in 100 g of tofu (14 g) and buckwheat groats (13 g). Protein is one of the key dietary macronutrients, with structural, building, metabolic, transport, immune and repair functions [43]. Proteins are subject to anabolic and catabolic processes. They are a component of the body’s tissues, hormones and cells. In addition, proteins were shown to have beneficial effects on wound healing, nutritional status and the prevention and treatment of malnutrition [44]. Physiological conditions with increased protein requirements include cancer, ageing, inflammation, heart failure, chronic obstructive pulmonary disease, chronic kidney disease, dialysis treatment, malnutrition and sport activity [45]. The dietary recommendations for adults developed by the National Centre for Nutrition Education (Narodowe Centrum Edukacji Żywieniowej) suggest the consumption of 0.9 g of protein per kilogram of body weight, while ESPEN’s dietary recommendations for the elderly indicate a protein consumption in the range from 1 g to 1.5 g per kilogram of body weight, in addition to individualised intakes depending on existing disease entities [45]. It is also important to note the current surge in popularity of meat-free diets, i.e., vegetarianism and veganism, which exclude or significantly restrict animal protein consumption. The search for a readily available source of diverse proteins is becoming essential, particularly for those following such diets. Matcha appears to be a very good nutritional choice to supplement dietary protein deficiencies. However, further research is needed, in particular to analyse the amino acid profile of different matcha types.

Koláčková T. et al. investigated the chemical composition, including fibre and protein content, of matcha tea [6]. The protein content ranged between 20.3% and 35.0%. The values at the lower end of the range were similar to those observed in our study (17.3%), while the upper end was more than 10% higher, showing that matcha green tea, depending on its quality, can be used to supplement the diet with plant protein. Wang J. et al. investigated dry green tea [46]. The results of their study in terms of the protein content of the product were very similar to ours (17.29%) and indicated a protein amount of 20.61 ± 0.23%. Both papers identified green tea powder as a promising source of protein.

In our study, total fat in dry matcha tea was 7.285 g/100 g, and we are among the first to examine this macronutrient. Fat is one of the basic nutrients and a source of energy. The recommended daily fat intake is individual and depends on age, energy requirements, comorbidities and individual metabolism. Fats provide energy and essential fatty acids (EFAs) and are involved in the absorption of vitamins A, D, E, K and of some bioactive compounds. Fats are also precursors to biologically active compounds in the body. An excessive dietary fat intake can have adverse health effects, including obesity, lipid abnormalities, cardiovascular disease and some malignant cancers [47].

Fats are divided into saturated fatty acids (SFAs) and unsaturated fatty acids, the latter being subdivided into monounsaturated fatty acids (MUFAs) and polyunsaturated fatty acids (PUFAs) [48]. Some polyunsaturated fatty acids are known as essential fatty acids, namely, linoleic acid and α-linolenic acid. MUFAs affect the serum triglyceride levels and improve glucose metabolism, in addition to being essential components of cell membranes. Omega-3 PUFAs (also known as n-3 PUFAs) are incorporated into the nervous tissue, improve insulin sensitivity, enhance peripheral glucose utilization, reduce obesity and have anti-atherosclerotic, anti-thrombotic and anti-inflammatory effects [49]. In turn, n-6 PUFAs reduce the concentration of LDL cholesterol. It is essential to maintain an adequate ratio of n-3 to n-6 PUFAs to maintain the body’s homeostasis [47].

The unsaturated fatty acid profile that we analysed for the first time in matcha showed that α-linolenic acid is the most abundant fatty acid in dry matcha, while palmitoleic acid of the omega-7 family is the least abundant. In terms of saturated fatty acids, there was a higher proportion of palmitic acid (15.4%) and a lower level of stearic acid (1.3%). When comparing the concentrations of unsaturated and saturated fatty acids in dry matcha, unsaturated fatty acids clearly predominated.

Alpha-linolenic acid (ALA), which was significantly predominant in matcha, is an n-3 PUFA of plant origin. Its consumption is correlated with a reduced risk of death from cardiovascular disease and ischaemic heart disease [50]. It has anti-inflammatory, immunomodulating [51], antidepressant and neuroprotective [52] properties. In addition, it accelerates wound healing and inhibits the proliferation of some cancers [53]. Linoleic acid (LA) is the most commonly consumed n-6 PUFA in Western plant-based diets [54,55]. LA may have metabolic effects on pancreatic beta-cell function and peripheral glucose uptake. Nowadays, linoleic acid tends to be consumed in excess, which interferes with the regulatory mechanisms of glucose metabolism by promoting insulin resistance [54]. Moreover, an excessive daily intake promotes inflammation [55]. When consumed in adequate amounts, it lowers total cholesterol [55] and is also essential for optimal inner mitochondrial membrane function [56]. A low LA intake correlates with reduced physical function in older people [56]. Oleic acid exhibits anti-inflammatory effects, making its use potentially beneficial in the prevention of breast cancer and rheumatoid arthritis by inhibiting pro-inflammatory mechanisms [57]. It protects the intestinal mucosa, preventing ulcers [57]. It was suggested that oleic acid may lower the blood pressure and reduce ROS [58]. Palmitoleic acid is found in the highest concentrations in donkey milk. It is also found in the milk of other mammals such as cows and goats [48]. It is synthesised from palmitic acid. It was shown to have health-promoting properties in mouse models and cell lines. It is able to regulate a variety of metabolic processes: it increases muscle insulin sensitivity and β-cell proliferation, prevents mitochondrial endoplasmic reticulum stress and exhibits lipogenic activity in white adipocytes [59]. Our study confirms the beneficial lipid profile of green tea powder, which may be responsible for matcha health-promoting properties.

Phenolic compounds belong to a class of secondary metabolites in plant products that exhibit high bioactivity, which can reduce the risk of onset and development of oxidative stress-related diseases [60]. Through the distinctive cultivation and preparation of matcha tea (shading the bushes and powdering the whole leaves), different contents of compounds, especially phenolic compounds, were expected [61,62]. During processing, the oxidation of tea phenols is reduced in green tea compared to other tea varieties [60]. This study is the first to examine the content of so many flavonoids and phenolic acids. The most abundant compound in matcha tea was caffeine—2213.492 µg/g—while among phenolic acids, gallic acid had the highest content—252.3755 µg/g. Chlorogenic acid, caffeic acid, dihydroxybenzoic acid and 2-hydroxycinnamic acid were not detected in the studied matcha tea powder. Caffeine is a potent antioxidant compound that is found in many beverages, including tea. In their study, Koláčková et al. determined caffeine levels in the range of 14.4–34.1 mg/g, which exceed the values obtained in this study [6]. Regarding the content of the second most abundant compound in this study, gallic acid, our results are comparable to those of the authors mentioned above, who reported a gallic acid content in the range of 39.4–423.0 μg/g. Epicatechin gallate is a catechin derivative, characteristic of tea, especially green tea. It is associated with potent antioxidant and anticancer effects [63]. The matcha tea in this study had a high content of this catechin (165.5024 µg/g), similar to the results of other authors [64]. The levels of compounds found in tea can vary due to different suppliers and geographical regions providing the tea, different harvesting times and, therefore, different lengths of time the leaves grow and mature [62].

The number of studies on the content of dietary fibre, protein, fat and, in particular, fatty acids in dry matcha tea is very limited. Therefore, given its rich chemical composition, it is worthwhile to continue and expand research on matcha and its versatile uses.

## 5. Conclusions

Japanese matcha tea powder is a valuable food product with a rich composition. It is rich in dietary fibre, especially insoluble fibre. Dry matcha was also shown to be a good source of plant protein. The most abundant unsaturated fatty acid in matcha powder was found to be α-linolenic acid, a member of the omega-3 family. A number of phenolic compounds were found, including particularly high levels of caffeine, gallic acid and epicatechin gallate. Due to its high concentration of nutrients and health-promoting compounds, matcha tea powder can be helpful in providing adequate amounts of the above-mentioned nutrients.

## Figures and Tables

**Figure 1 foods-13-01167-f001:**
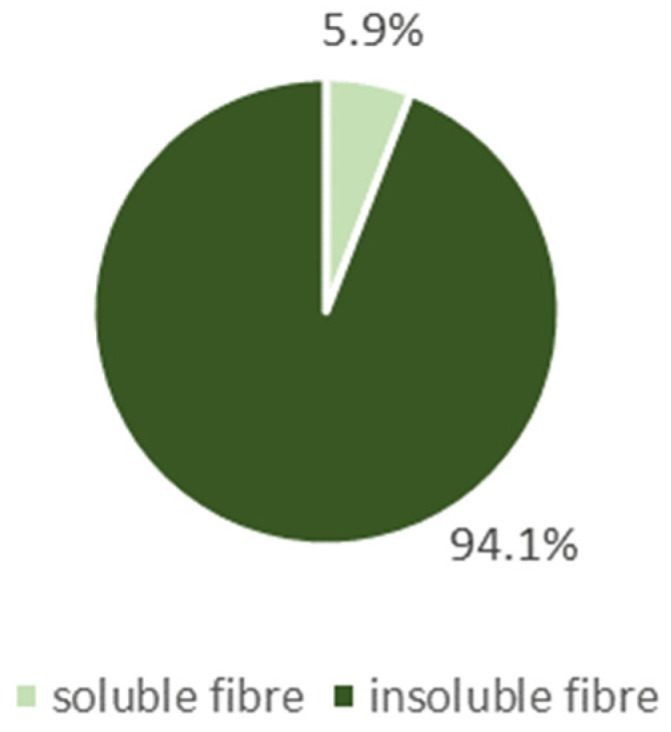
Content of dietary fibre fractions [%].

**Figure 2 foods-13-01167-f002:**
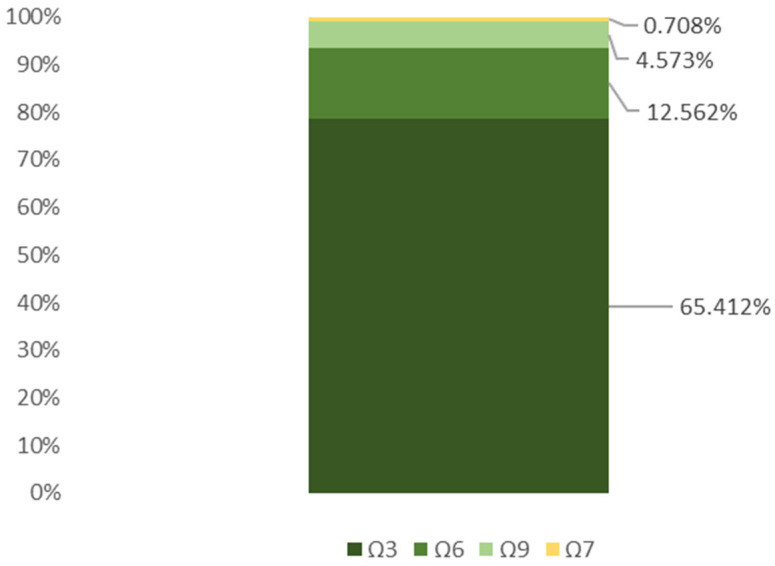
Proportions of omega fatty acid families in total fat content [%].

**Table 1 foods-13-01167-t001:** Macronutrient and element content of matcha tea powder.

	Unit	Mean ± SD
protein	g/100 g	17.287 ± 1.636
nitrogen	g/100 g	3.012 ± 0.477
total dietary fibre	g/100 g	56.130 ± 2.753
soluble fibre	g/100 g	3.3 ± 0.141
insoluble fibre	g/100 g	52.830 ± 2.611
total fat	g/100 g	7.285 ± 0.006

**Table 2 foods-13-01167-t002:** Percentage content of unsaturated fatty acids in matcha.

Regular Name	Systematic Name	Numeric Symbol	Omega Family	Semi-Structural Formula	Mean	SD
palmitoleic acid	(Z)-9-hexadekaenoic acid	C 16:1	7	C_16_H_30_O_2_	0.206	0.021
oleic acid	(Z)-9-octadecenoic acid	C 18:1	9	C_18_H_34_O_2_	4.573	0.037
vaccenic acid	(E)-11-octadecenoic acid	C 18:1	7	C_18_H_34_O_2_	0.502	0.016
linoleic acid	(Z,Z)-9,12-octadecadienoic acid	C 18:2	6	C_18_H_32_O_2_	12.562	0.158
α-linolenic acid	(Z,Z,Z)-9,12,15-octadecatrienoic acid	C 18:3	3	C_18_H_30_O_2_	65.412	0.453

**Table 3 foods-13-01167-t003:** Percentage content of saturated fatty acids in matcha.

Regular Name	Systematic Name	Numeric Symbol	Semi-Structural Formula	Mean	SD
palmitic acid	hexadecanoic acid	C 16:0	CH_3_(CH_2_)_14_COOH	15.418	0.233
stearic acid	octadecanoic acid	C 18:0	CH_3_(CH_2_)_17_COOH	1.327	0.011

**Table 4 foods-13-01167-t004:** Phenolic compound and caffeine content in matcha.

Phenolic Compounds	Mean (µg/g) ± SD
gallic acid	252.3755 ± 12.82054
dihydroxybenzoic acid	0 ± 0
4-hydroxybenzoic acid	41.83232 ± 2.125059
chlorogenic acid	0 ± 0
caffeic acid	0 ± 0
caffeine	2213.492 ± 112.4442
p-coumaric	3.265168 ± 0.165869
ferulic acid	15.51277 ± 0.78804
epicatechin gallate	165.5024 ± 8.407435
sinapic acid	29.15353 ± 1.480984
ellagic acid	10.82998 ± 0.550157
2-hydroxycinnamic acid	0 ± 0
rutin	35.34143 ± 1.795326
resveratrol	14.47955 ± 0.735553
myricetin	108.1979 ± 5.496397
quercetin	5.279351 ± 0.268188
kaempferol	1.462069 ± 0.074272
apigenin	8.304355 ± 0.421857

## Data Availability

The original contributions presented in the study are included in the article, further inquiries can be directed to the corresponding author.

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
