# Peer review of "Matcha Green Tea: Chemical Composition, Phenolic Acids, Caffeine and Fatty Acid Profile"

_foods, 2024, doi:10.3390/foods13081167_

Round 1

Reviewer 1 Report

Comments and Suggestions for Authors

1.        The title is too long, choose concise language to highlight the purpose of the study.

2.        The authors analyzed the fiber, fatty acids and phenolic acid components of matcha, but the samples were small, so how can you ensure that the study results have universality?

3.        The description of fatty acids uses an excessive number of tables and figures, which is too redundant with reading and comprehension, and it is recommended that they be consolidated into a maximum of 1-2 tables and figures.

4.        Lines 155-156 and lines156-158 are repeated.

5.        In the manuscript, the authors mention that matcha is particularly high in total dietary fiber, with a predominance of insoluble components. However, in the Introduction, matcha is described as a green tea beverage, is its insoluble content not valuable? The authors also mention in conclusions that matcha powder is a valuable food product rich in ingredients, is it possible to introduce the common ways of utilizing matcha in the introduction and analyze them appropriately in the discussion?

6.        Discussion is too long and should avoid listing the references, suggesting that it be analyzed in conjunction with results.

7.        The conclusion should focus on the results of the study, not the discussion.

Comments on the Quality of English Language

Generally

Reviewer 2 Report

Comments and Suggestions for Authors

This research contribution by Bika et al., was based on determination of the nutritional composition and fatty acid profile of Japanese green tea (Matcha) by proximate and GC analysis, repectively. The authors should be commended for an effort to provide some basic insights with respect to the chemical composition and nutritional value of Matcha powder. This could be beneficial to individuals who consume the powder and food product developers. That said, the manuscript is seriously deficient in many ways. Firstly, it is not hypothesis-driven and does not attempt to answer any research question. Determining the composition of a food product does not constitute a research, it is food product analysis. In fact, the points raised in the Conclusions section pertaining the potential application of the Matcha powder could provide some clues in designing a robust hypothesis-based research. Other minor points worthy of considertaion are provided below.

-Title is too long. Try to make it simple and captivating. A good rule of thumb is not more than twenty words. It should be concise and descriptive to allow it to be easily discovered by readers.

-Section 2.2. was the tea material preserved prior to analysis?

-Analytical standard is not the name of a reagent. Please replace that with the name of an actual reagent.

-Section 2.3. Please wen you described materials and methods, ensure that basic details such as amount or centration of reagents, volume, time, temperature, etc. are included in a step-by-step manner to ensure reproducibility/replicability of the technique. This should be applicable in all the methods described. Therefore, a meticulous revision of the methods section is strongly recommended.

-Section 2.5. Please include the detection method for the GC. What was it coupled to in order for the separated analytes to be detected?

-Why analyze the powder when Matcha is often consumed in the form of decoction prepared from the powder. Wouldn’t the extract be more appropriate for analysis?

Comments on the Quality of English Language

The manuscript is replete with grammatical errors and should be thoroughly edited for English language.

Reviewer 3 Report

Comments and Suggestions for Authors

In this research the authors evaluated the chemical composition of matcha-powdered Japanese green tea. I suggest the authors consider the following suggestions for further assessment before publication. But my main concern revolves around the distinctive aspects (uniqueness) of this research.

·      The title of the manuscript is unnecessarily long. It can be “Chemical Composition of Matcha-powdered Japanese Green Tea”

·      What is the uniqueness of your study? There are various other studies in the literature that provided information about the chemical composition of Matcha tea.

·      Lines 84-87: Who provided you Matcha tea or did you buy them? More information is needed.

·      Lines 84-87: How many or how much sample you obtained, if it is only from one location of Matcha tea, don’t you think the location, etc. would affect the chemical composition of the Matcha tea.

·      Lines 155 – 158: The same sentence has been duplicated.

·      Figure 1: That figure is unnecessary, you have already stated that info in the text (Lines 159-162).

·      Similar to the previous comment, Figure 2 is unnecessary.

·      In the keywords, the authors have mentioned “prebiotics”, but they talked about prebiotics in only one sentence. If it is such an important attribute and would like to use it as a keyword, please include more info on the prebiotics from literature.

Comments on the Quality of English Language

There are just minor English language problems.

Round 2

Reviewer 1 Report

Comments and Suggestions for Authors

It's been revised as requested.

Reviewer 2 Report

Comments and Suggestions for Authors

The substantial issues raised in the the reviewers' comments had not been satisfactorily addressed by the authors.

Thank you. 

Reviewer 3 Report

Comments and Suggestions for Authors

The authors have made the necessary revisions to the manuscript in line with the suggestions provided. However, in the initial round of corrections, I asked the authors to elucidate the uniqueness of their study, and they provided a good explanation. I recommend that they integrate this explanation into the manuscript. By incorporating this insight, readers will gain a clearer understanding of the novel aspects and significance of the study.

Comments on the Quality of English Language

The English language is fine, just minor typos.
